# Adjuvant Chemoradiotherapy or Radiotherapy Alone for Early Squamous Cervical Cancer with a Single Surgical-Pathological High-Risk Factor

**DOI:** 10.3390/cancers17122041

**Published:** 2025-06-18

**Authors:** Ester P. Olthof, Hans H. B. Wenzel, Jacobus van der Velden, Lukas J. A. Stalpers, Maaike A. van der Aa, Constantijne H. Mom

**Affiliations:** 1Department of Research & Development, Netherlands Comprehensive Cancer Organisation, 3511 CV Utrecht, The Netherlands; 2Centre for Gynaecologic Oncology Amsterdam (CGOA), Department of Gynaecological Oncology, Amsterdam University Medical Centre, 1081 HV Amsterdam, The Netherlands; 3Department of Radiation Oncology, Amsterdam University Medical Centre, 1081 HV Amsterdam, The Netherlands

**Keywords:** uterine cervical cancer, adjuvant chemoradiotherapy, adjuvant radiotherapy

## Abstract

Adjuvant chemoradiotherapy following radical hysterectomy has been shown to be effective in improving survival rates for patients with early-stage squamous cervical cancer and high-risk features following surgery. However, emerging data suggest that certain subgroups may not benefit from it and may experience unnecessary toxicity and overtreatment. This study analysed 122 patients with squamous cell carcinoma and a single high-risk factor—positive resection margins, parametrial involvement, or pelvic lymph node metastasis. Of these patients, 76 (62%) received adjuvant chemoradiotherapy, while 46 (38%) received adjuvant radiotherapy alone. Inverse probability of treatment weighting (IPTW) was used to adjust for confounding factors. Five-year recurrence-free survival and overall survival rates were comparable between the adjuvant chemoradiotherapy and radiotherapy groups (84% versus 91%; *p* = 0.49). Our results suggest that adding chemotherapy may not improve survival in this patient group. These findings support a more individualised adjuvant treatment strategy to avoid unnecessary toxicity while maintaining oncological efficacy.

## 1. Introduction

Radical hysterectomy with pelvic lymphadenectomy is the standard of care for patients with International Federation of Gynaecology and Obstetrics (FIGO) 2009 stage IA2-IIA2 cervical cancer [1,2]. The presence of surgical pathological factors have been associated with a high risk of recurrence and poor survival [1,2]. These high-risk factors include tumour-positive resection margins, parametrial involvement, and lymph node metastases. For patients with one or more high-risk factors, adjuvant pelvic external beam radiotherapy (i.e., 45–50 Gy) and concurrent chemotherapy (i.e., cisplatin 40 mg/m^2^ weekly), known as chemoradiotherapy, is recommended [1,2]. The presence of parametrial involvement or lymph node metastases results in upstaging to FIGO (2018) stage IIB and IIIC, respectively [3]. Both of these stages are classified as locally advanced cervical cancer, for which the standard treatment is primary chemoradiotherapy.

Support for the recommendation of adjuvant chemoradiotherapy in cases with high-risk factors comes from a randomised controlled trial comparing adjuvant chemoradiotherapy with radiotherapy in 268 patients with early-stage cervical cancer who had undergone surgery and had high-risk factors after the operation. Four-year recurrence-free survival (80% versus 63%) and overall survival (81% versus 71%) were superior after adjuvant chemoradiotherapy, albeit at the cost of increased grade 3 and 4 haematological and gastrointestinal toxicities [4]. However, post hoc analysis revealed minimal or no survival benefit for certain subgroups, e.g., patients with tumours ≤2 cm or with only one nodal metastasis [5].

In addition, an earlier study by Samlal et al. (1997) described a group with positive nodes, and a squamous histology without parametrial invasion, whose 5-year disease-specific survival was significantly better than those with positive nodes and non-squamous histology or parametrial invasion (94% versus 60%). All patients were treated with postoperative radiotherapy without chemotherapy [6]. These results are supported by a more recent study, describing that the clinical outcome after adjuvant chemoradiotherapy is associated with the number of high-risk factors [7].

Based on these studies, the addition of concurrent chemotherapy to adjuvant radiotherapy may be questioned in a subset of patients with a single high-risk surgical-pathological factor. Therefore, the aim of this retrospective study is to compare the benefit of adjuvant chemoradiotherapy with that of radiotherapy in patients with clinically early-stage squamous cell cancer of the cervix treated with radical hysterectomy and pelvic lymphadenectomy, and a single postoperative high-risk factor, in terms of recurrence-free survival and overall survival.

## 2. Materials and Methods

### 2.1. Study Design and Data Collection

In this retrospective cohort study, we compared patients treated for cervical cancer at the Amsterdam University Medical Centre (Amsterdam UMC), selected from the Amsterdam UMC hospital database, with patients treated at the other Dutch oncological centres, selected from the population-based Netherlands Cancer Registry (NCR). This study was approved by the Privacy Review Board of the NCR (No K23.218). With regard to the adjuvant treatment of patients with squamous cell cancer and one high-risk factor, the Amsterdam UMC deviates from the Dutch national cervical cancer guideline, as these patients are treated with adjuvant radiotherapy only instead of chemoradiotherapy, as recommended in the guideline [2].

The inclusion criteria were as follows: (1) squamous cell carcinoma of the cervix, (2) diagnosis in ≥2001, (3) clinical tumour stage cT IA2-IIA2, (4) treatment with radical hysterectomy and pelvic (±para-aortic) lymphadenectomy followed by adjuvant chemoradiotherapy or radiotherapy, and (5) presence of only one postoperatively confirmed high-risk factor: positive resection margins, parametrial involvement, or pelvic lymph node metastases. Patients receiving neoadjuvant chemotherapy were excluded. A positive resection margin was defined as the presence of tumour cells in the surgical margin on histological examination. Adjuvant radiotherapy consisted of pelvic external beam radiotherapy (i.e., 45–50 Gy), and in the case of concurrent chemotherapy, cisplatin 40 mg/m^2^ weekly was added, according to the current guidelines [1,2]. Patient, tumour, and treatment characteristics were extracted from both databases.

### 2.2. Outcomes and Definitions

Recurrence-free survival and overall survival are the primary outcomes of this study. The first was defined as the interval from the date of surgery to recurrence status, which was determined from hospital records. We censored patients who were alive without recurrence or lost to follow-up at the date of enrolment or the last date of clinical contact, respectively. Overall survival was calculated from diagnosis to death. Obtaining the vital status in the NCR database was achieved through linkage to the Municipal Personal Records Database (updated to 31 January 2023). Patients who were still alive on this date were censored.

### 2.3. Statistical Analysis

The Mann–Whitney U, Kruskal–Wallis, and Fisher’s exact tests were used to perform descriptive statistics, while the Kaplan−Meier method, log-rank test, and multivariable Cox regression (where appropriate) were used to perform unadjusted survival analyses. To adjust for measured confounding between treatment groups, propensity score analysis was employed. Propensity scores were estimated using a logistic regression model incorporating variables associated with the outcome or both the outcome and treatment, including age, lymph node metastasis, parametrial invasion, clinical tumour size, and lymphovascular space invasion. FIGO stage was excluded due to overlap with included variables, while surgical resection margin status was omitted due to multicollinearity.

Inverse probability of treatment weighting (IPTW) was applied within Cox proportional hazard models to account for confounding and achieve balance between treatment groups in the survival analyses. This approach was selected due to the limited number of events and the relatively small sample size, which rendered conventional multivariable adjustment less suitable despite the need to control for baseline differences. Covariate balance was assessed using absolute standardised differences, with values of ≤0.10 indicating adequate balance [8]. To assess the potential impact of the surgical approach on survival, as highlighted in the LACC trial, a sensitivity analysis was conducted, including only patients who underwent an open hysterectomy [9]. In addition, a subgroup analysis was performed for patients with pelvic lymph node metastasis, as this was the majority group. All statistical analyses were performed using Stata version 17.0 (StataCorp, College Station, TX, USA), with a *p*-value of <0.05 being considered statistically significant.

## 3. Results

### 3.1. Baseline Characteristics

Adjuvant chemoradiotherapy was administered to 106/750 (12%) of patients with early-stage squamous cell cervical cancer, treated by radical hysterectomy and selected from the national database. Of which 76/106 (72%) of patients had a single high-risk factor. Additionally, adjuvant radiotherapy was administered to 114/389 (29%) of patients from the single centre database, of which 45/114 (39%) had a single high-risk factor—122 patients with one high-risk factor after radical hysterectomy and pelvic lymphadenectomy, 76 (62%) received adjuvant chemoradiotherapy, and 46 (38%) received adjuvant radiotherapy. Baseline characteristics are shown in Table 1. Pathological parametrial invasion was observed in 7% of the chemoradiotherapy group and 20% of the radiotherapy alone group (*p* = 0.040). Positive resection margins were observed in 4% of both groups (*p* = 1.00). Pelvic lymph node metastases were found in 89% of the chemoradiotherapy group and 76% of the radiotherapy alone group (*p* = 0.07). Poor prognostic features, such as larger tumour size (35 mm versus 29 mm; *p* = 0.021), tumour grade 3 (82% versus 48%; *p* < 0.001), and pathological parametrial invasion (20% versus 7%; *p* = 0.040) were more common in the adjuvant radiotherapy group than in the chemoradiotherapy group. However, patients who received adjuvant chemoradiotherapy had more pelvic lymph node metastases (mean 1.8 versus 1.3; *p* = 0.025). All propensity score model characteristics (i.e., clinical tumour size, pathological parametrial invasion, lymph node metastasis, and lymphovascular space invasion) were balanced after inverse probability treatment weighting, with a standardized mean difference of ≤0.10 (Appendix A).

### 3.2. Survival

The median follow-up registration for recurrence-free survival (9 versus 5 years; *p* < 0.001) and overall survival (8 versus 6 years; *p* < 0.001) was found to be longer in the adjuvant chemoradiotherapy group compared with the adjuvant radiotherapy group (Table 1). Recurrence was observed in 13 patients (17%) after adjuvant chemoradiotherapy and in five patients (11%) after adjuvant radiotherapy (*p* = 0.44), with distant metastases (54–80%) being the most common side of failure. Mortality rates were not significantly different between the two treatment groups, with 20% versus 11% deaths after adjuvant chemoradiotherapy versus radiotherapy (*p* = 0.31).

The unadjusted and adjusted 5-year survival rates were comparable between the adjuvant chemoradiotherapy and radiotherapy groups for both recurrence-free survival (85% versus 87%; *p* = 0.58 and 84% versus 91%; *p* = 0.29) and overall survival (84% versus 87%; *p* = 0.61 and 84% versus 91%; *p* = 0.30); see Figure 1 and Figure 2, respectively. As shown in Table 2, type of adjuvant therapy was not associated with recurrence-free survival (radiotherapy HR 0.54; 95% confidence interval 0.17–1.71; *p* = 0.29) or overall survival (radiotherapy HR 0.56; 0.19–1.67; *p* = 0.30) after adjustment for confounding by inverse probability treatment weighting. An analysis adjusted for confounding by site of recurrence showed a decreased risk of locoregional recurrence after adjuvant radiotherapy (HR 0.09; 0.01–0.79; *p* = 0.03), while the risk of distant recurrence was almost the same as after adjuvant chemoradiotherapy (HR 0.85; 0.21–3.37; *p* = 0.81); see Appendix A.

Sensitivity analysis in patients who underwent open surgery showed comparable results to the primary analysis for recurrence-free survival (radiotherapy HR 0.58; 0.18–1.87; *p* = 0.37) and overall survival (radiotherapy HR 0.55; 0.19–1.63; *p* = 0.28). In addition, minimally invasive surgery was not negatively associated with recurrence-free survival (HR 0.74; 0.29–1.87; *p* = 0.52) or overall survival (HR 0.64; 95% CI 0.25–1.61; *p* = 0.34) when compared with open surgery on univariate analysis; see Appendix A. Unadjusted subgroup analysis of patients with pelvic lymph node metastases demonstrated that radiotherapy alone did not lead to a decrease in either recurrence-free or overall survival (HR 0.42; 95% CI 0.12–1.53; *p* = 0.19 and HR 0.51; 95% CI 0.15–1.82; *p* = 0.30, respectively).

## 4. Discussion

This study compared the efficacy of chemoradiotherapy versus radiotherapy alone as adjuvant treatment strategies for women diagnosed with clinically early-stage cervical squamous cell carcinoma and one high-risk surgical pathological factor (i.e., positive resection margins, parametrial involvement or pelvic lymph node metastases) after radical hysterectomy and pelvic lymphadenectomy. We found no superiority for the addition of chemotherapy to adjuvant radiotherapy in terms of recurrence-free survival and overall survival. For clinical practice these results suggest that patients with one single high-risk factor may currently be over-treated if guidelines are strictly followed.

The guideline recommendation for adjuvant chemoradiotherapy in patients with high-risk factors is based on Level I evidence derived from a randomised controlled trial (RCT) conducted in the 1990s by Peters et al [4]. In contrast, our results do not show a survival benefit of adding chemotherapy to adjuvant radiotherapy. The 5-year survival rates in our two groups (all ≥ 84%) are comparable to those in the RCT after chemoradiotherapy (≥80%), but significantly higher than those in the RCT after radiotherapy (≥63%) [4]. This discrepancy may be due to differences in inclusion criteria between our study and the RCT. Our study included a relatively favourable group of patients, selecting only those with squamous cell carcinoma and a single high-risk factor, whereas the RCT included patients with non-squamous cell carcinoma and multiple high-risk factors. Furthermore, the radiotherapy group in the Peters et al. study had a higher rate of parametrial involvement (35% versus 20% in our study) and a higher rate of positive pelvic nodes (84% versus 76%).

Although we did not find an association between recurrence-free survival and the type of adjuvant therapy, we did find a reduced risk of locoregional recurrence in the group of patients who received adjuvant radiotherapy without chemotherapy after adjustment for confounders. However, we believe that this association is not related to the type of adjuvant therapy, as both groups received radiotherapy [10]. We assume that this observation may be due to the radicality of the hysterectomy. All patients who received adjuvant radiotherapy underwent a radical hysterectomy according to Querleu−Morrow type C2, whereas the radicality of hysterectomy in patients in the chemoradiotherapy group varied between type A-C2 according to guidelines and local practice [1,11]. A more radical hysterectomy is associated with better disease-free survival for tumours >20 mm according to Derks et al. (2017) [12]. Notably, whether adjuvant radiotherapy may be sufficient after type A-C1 hysterectomy cannot be answered based on our data. Another factor that may have influenced our locoregional recurrence results is the surgical approach, as 21% (compared to 0%) of patients in the chemoradiotherapy group were treated with minimally invasive surgery, which has been shown to be associated with a higher frequency of locoregional recurrence according to the LACC trial [9].

The concept of an adjuvant treatment approach without chemotherapy was distilled from Samlal et al. (1997) and Monk et al. (2005) [5,6]. Samlal and colleagues identified a low-risk subgroup of surgically treated node positive squamous cell carcinoma patients without parametrial invasion [6]. This group received adjuvant radiotherapy only and had a 5-year survival rate of 94%. The study by Monk et al. was a re-analysis of the RCT by Peters et al. (2000) [4,5]. They showed a survival benefit after adding chemotherapy to adjuvant radiotherapy in patients with poor prognostic factors (i.e., non-squamous histological subtype, presence of parametrial invasion, ≥2 lymph node metastases and tumours ≥2 cm). These results suggest that other combinations of prognostic factors may also represent subgroups for which adjuvant radiotherapy may be sufficient, such as tumours ≤2 cm with one high-risk factor.

In line with our findings, previous studies on postoperative adjuvant therapy have reported comparable survival rates between chemoradiotherapy and radiotherapy alone [13,14]. However, these studies reported lower rates of recurrence-free survival (65–66%) and cervical cancer mortality (73–78%) than ours. This may be attributed to the inclusion of patients with poorer prognostic profiles: node-positive patients with non-squamous histological subtypes (15–51%) and para-aortic lymph node metastases and parametrial invasion (35–44%). Our study included only patients with squamous cell carcinoma and a single pathological high-risk factor, representing a more favourable subgroup likely to be adequately treated with radiotherapy alone. This limits comparability with other studies on adjuvant therapy in broader populations. For example, a randomised trial of 1048 patients comparing sequential chemoradiotherapy, concurrent chemoradiotherapy, and radiotherapy alone found no significant disease-free or overall survival benefit for concurrent chemoradiotherapy over radiotherapy (HR 0.79; 95% CI 0.56–1.12; *p* = 0.19 and HR 0.78; 95% CI 0.49–1.23; *p* = 0.29, respectively) [15]. Sequential chemoradiotherapy, however, showed improved outcomes (DFS compared with radiotherapy HR 0.52; 95% CI 0.35–0.76; *p* = 0.001 and compared with concurrent chemoradiotherapy HR 0.65; 95% CI 0.44–0.96; *p* = 0.03). Yet, their cohort differed substantially, including patients with deep stromal invasion (83–86%) and potentially >1 pathological adverse risk factor. As an alternative to (chemo-)radiotherapy, chemotherapy alone is currently being investigated as adjuvant therapy in a clinical trial for patients with high-risk factors following radical hysterectomy and pelvic lymphadenectomy [16]. Retrospective studies have shown the potential efficacy of chemotherapy as a postoperative treatment strategy for selected subgroups of node-positive patients [13,17].

This study represents a pioneering exploration of adjuvant strategies specifically tailored to patients with early-stage squamous cell carcinoma and either positive resection margins, parametrial involvement, or pelvic lymph node metastases, with criteria selected on the basis of evidence [5,6]. Furthermore, the choice of adjuvant therapy strategy in this study, either chemoradiotherapy or radiotherapy, was determined by the treatment centre, independent of potential confounders such as age and comorbidities, which could potentially influence the results.

Several limitations must be discussed. First, the heterogeneity of prognostic factors between our treatment groups (i.e., tumour diameter, tumour grade, parametrial invasion, number of lymph node metastases, and surgical approach) may have biased our results, although we attempted to adjust for most of these factors by inverse probability treatment weighting and by performing sensitivity analyses. Second, our study may suffer from underpowering due to our reliance on a small cohort, a consequence of our deliberate selection of a specific patient population. Third, the lack of data due to our retrospective study design, including depth of invasion and details of radiotherapy. In particular, data on toxicity and quality of life are a notable limitation. Given the already high survival rates in early-stage cervical cancer, these outcomes are important factors in treatment selection. The literature suggests that radiotherapy may be better tolerated than chemoradiotherapy and may, therefore, be associated with a better quality of life [18]. Future research could focus on the development and validation of nomograms to predict the risk of recurrence after surgery, which would allow for more precise determination of optimal adjuvant therapy strategies [19,20,21].

## 5. Conclusions

In conclusion, adjuvant radiotherapy instead of chemoradiotherapy may be sufficient for women diagnosed with early-stage cervical squamous cell carcinoma and one high-risk surgical pathological factor (i.e., positive resection margins, parametrial involvement, or pelvic lymph node metastasis) after radical hysterectomy and pelvic lymphadenectomy. However, caution should be exercised in interpreting these findings due to the limitations of our small and heterogeneous study cohort. Future research to validate our findings in larger prospective cohorts is warranted to reduce chemotherapy-related toxicity in this specific population.

## Figures and Tables

**Figure 1 cancers-17-02041-f001:**
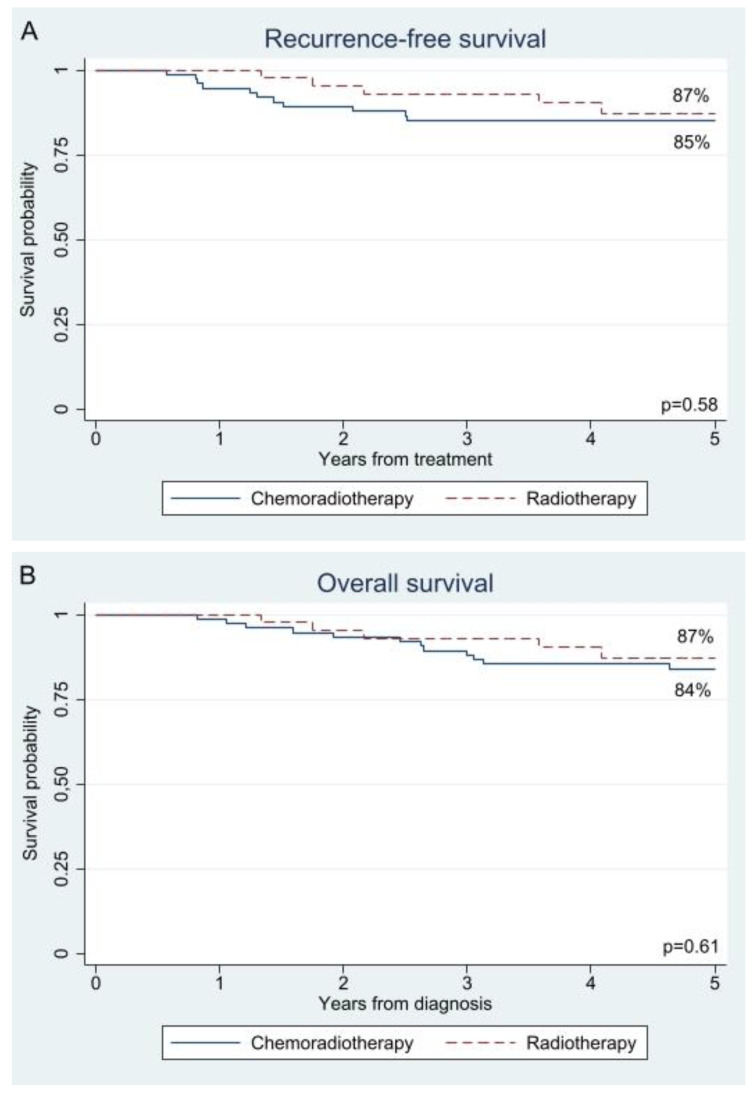
Kaplan−Meier curves for (**A**) recurrence-free survival and (**B**) overall survival in patients who treated with adjuvant radiotherapy or chemoradiotherapy.

**Figure 2 cancers-17-02041-f002:**
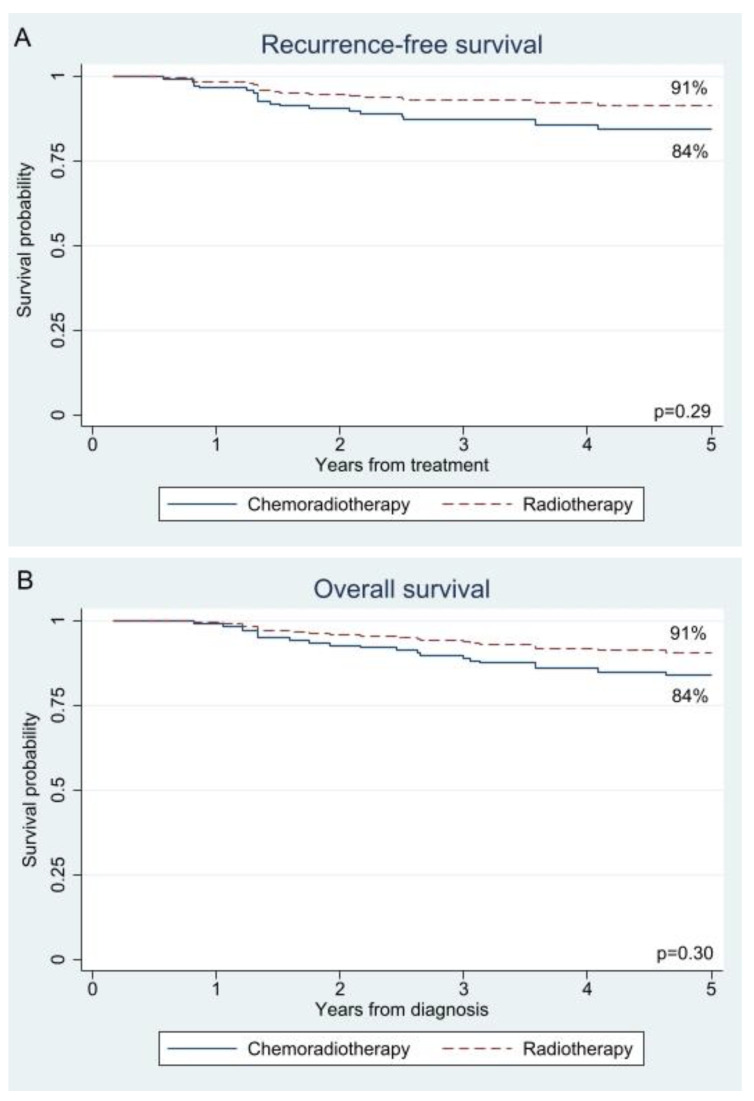
For confounding adjusted survival curves for (**A**) recurrence-free survival and (**B**) overall survival in patients who were treated with adjuvant radiotherapy or chemoradiotherapy.

**Table 1 cancers-17-02041-t001:** Baseline characteristics according to the type of adjuvant therapy.

Characteristics	Missing	Chemoradiotherapy	Radiotherapy	*p*-Value
N	0 (0%)	76 (62%)	46 (38%)	-
Median age (range)	0 (0%)	42 (28–74)	45 (26–71)	0.26
Year of diagnosis (range)	0 (0%)	2012 (2009–2017)	2010 (2001–2018)	0.008 §
FIGO 2018 *	0 (0%)			0.06
IB		3 (4%)	1 (2%)	
IIA		0 (0%)	1 (2%)	
IIB		5 (7%)	9 (20%)	
IIIC1		68 (89%)	35 (76%)	
Mean tumour diameter in mm (standard deviation)	1 (1%)	29 (12.6)	35 (16.8)	0.021 §
Lymphovascular space invasion, yes	2 (2%)	59 (80%)	30 (65%)	0.09
Tumour grade	11 (9%)			<0.001 §
1		1 (2%)	0 (0%)	
2		33 (50%)	8 (18%)	
3		32 (48%)	37 (82%)	
Pathological parametrial invasion	0 (0%)	5 (7%)	9 (20%)	0.040 §
Positive resection margin	0 (0%)	3 (4%)	2 (4%)	1.00
Lymph node metastasis	0 (0%)	68 (89%)	35 (76%)	0.07
Median number of lymph node metastasis (range)	0 (0%)	1 (1–5)	1 (1–4)	0.025 §
Mean number of lymph node metastasis (standard deviation) †	0 (0%)	1.82 (1.17)	1.34 (0.73)	
Surgical approach	0 (0%)			<0.001 §
Open		60 (79%)	46 (100%)	
Minimally invasive		16 (21%)	0 (0%)	
Oncological outcome				
Median follow-up time for recurrences in years (range)	0 (0%)	9 (0–14)	5 (0–17)	<0.001 §
Mean follow-up time for overall survival in years (standard deviation)	0 (0%)	8 (3.6)	6 (3.7)	<0.001 §
Vital status, death	0 (0%)	15 (20%)	5 (11%)	0.31
Recurrence, yes	0 (0%)	13 (17%)	5 (11%)	0.44
Recurrence site ‡	0 (0%)			0.71
Locoregionaal		5 (38%)	1 (20%)	
Para-aortic		1 (8%)	0 (0%)	
Distant		7 (54%)	4 (80%)	

Abbreviations: FIGO, International Federation of Gynaecology and Obstetrics. * Converted from the FIGO 2009 and postoperatively confirmed. † For patients with a positive lymph node status only. ‡ For patients with a recurrence only. § Statistically significant.

**Table 2 cancers-17-02041-t002:** Cox regression survival analyses.

Analysis Type		Recurrence-Free Survival			Overall Survival		
		HR	95% CI	*p*-Value	HR	95% CI	*p*-Value
Univariable	CRT	1.00	Reference		1.00	Reference	
	RT	0.59	0.21–1.65	0.31	0.68	0.24–1.88	0.46
IPTW	CRT	1.00	Reference		1.00	Reference	
	RT	0.54	0.17–1.71	0.29	0.56	0.19–1.67	0.30

Abbreviations: IPTW, inverse probability treatment weighting; RT, radiotherapy; CRT chemoradiotherapy; HR, hazard ratio; CI, confidence interval.

## Data Availability

Data are contained within the article and Appendix A.

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
