# Peer review of "Adjuvant Chemoradiotherapy or Radiotherapy Alone for Early Squamous Cervical Cancer with a Single Surgical-Pathological High-Risk Factor"

_cancers, 2025, doi:10.3390/cancers17122041_

Round 1
Reviewer 1 Report
Comments and Suggestions for Authors
Adjuvant chemoradiotherapy or radiotherapy alone for early squamous cervical cancer with a single surgical-pathological high-risk factor Authors Ester P. Olthof et al
Study concerned 122 patients with squamous cell carcinoma treated with adjuvant chemoradiotherapy or radiotherapy alone and concluded by not improving patient survival. The authors have well described the details of patient inclusion, but the overall results could be compared with other studies. The report is not well written, a simple summary is missing (line 5) in the methodology, (ligne 25) the inclusion-exclusion criteria are missing; consequently, this work cannot be accepted for publication.
Author Response
We would also like to thank you for the very valuable and relevant comments and questions. We have managed to address all the comments, see attachment. If we have not addressed all the comments satisfactorily, we are willing to make additional changes.
Best wishes,

Reviewer 2 Report
Comments and Suggestions for Authors
Dear Authors,
You have taken up an interesting, little-explored topic of adjuvant treatment (RT vs chemoRT) in patients operated on for early cervical cancer with a single surgical-pathological high-risk factor. The analysis conducted has shown that in the analyzed group of patients, both methods of adjuvant treatment give similar results, which may contribute to changing the opinion on de-escalation of postoperative treatment in selected clinical situations.
I hope that the proposal to consider several of the following suggestions may prove helpful.
Part Simple Summary need to be completed
Abstract: I propose to replace the beginning of the 1st sentence (lines 22-24) with: This study aims to explore the benefit of adjuvant chemoradiotherapy compare with radiotherapy alone in patients …..
Results: At the beginning of this part, I propose to provide information on what percentage of all pts with IA2-IIA2 stage and operated on in the analyzed period were those with single high risk factor.
Table 1: The table shows that the largest group consisted of pts with affected pelvic nodes (103/122 pts), 68 pts in the radiochemotherapy group and 35 pts in the radiotherapy group. I propose to consider comparing the results of adjuvant therapy in both groups.
Discussion:
What determined the qualification for RT or chtRT ? Was it dictated by the type of high-risk factor? Although the authors briefly mention that … Furthermore, the choice of adjuvant therapy strategy in this study, either chemoradiotherapy or radiotherapy, was determined by the treatment center (lines 254-255), but perhaps more detailed information on this topic is available
Author Response
We would also like to thank you for the very valuable and relevant comments and questions. We have managed to address all the comments, see attachment. If we have not addressed all the comments satisfactorily, we are willing to make additional changes.
Kind regards,

Reviewer 3 Report
Comments and Suggestions for Authors
The authors present an interesting retrospective study about the benefit of the addition of chemotherapy to radiotherapy in the adjuvant setting of cervical cancer with one single high-risk factor.
The manuscript is well written, there are only a few comments:
The simple summary should be added as the manuscript includes the template
Authors should discuss the FIGO upstaging of parametrial infoltration and lymph node metastases to FIGO IIb and III
Authors should provide more data on the applied RT: How was RT delivered- IMRT or 3D? Was a boost applied to residual tumor or lymph node metastases? Was a brachyboost applied to the vaginal cuff?
The discussion section should be revised intensively and discuss other studies such as the STARS trial (He Huang et al. JAMA Oncol. 2021.)…
Author Response

(The authors gave the same response as above.)

Round 2
Reviewer 3 Report
Comments and Suggestions for Authors
Authors have revised the manuscript and addressed all concerns.